# Life Satisfaction, Courage, and Career Adaptability in a Group of Italian Workers

Sara Santilli *, Isabella Valbusa *, Barbara Rinaldi and Maria Cristina Ginevra

Department of Philosophy, Sociology, Pedagogy and Applied Psychology, University of Padova, Via Venezia 14, 35131 Padova, Italy; barbara.rinaldi@unipd.it (B.R.); mariacristina.ginevra@unipd.it (M.C.G.)
* Correspondence: sara.santilli@unipd.it (S.S.); isabella.valbusa@phd.unipd.it (I.V.)

**Abstract:** Today's work market is both unsteady and unpredictable, and this requires taking urgent and practical actions aiming at creating work opportunities and "better" jobs, promoting a social and solidarity economy, and encouraging the development of moral strength in the workplace. From the Life Design approach perspective, our study examines two variables necessary to cope with the current labor market, courage, and career adaptability, and their role in life satisfaction. Through courage, a full mediational model between life satisfaction and career adaptability was tested in the 525 (291 men and 234 women) employees involved in the present study. Results support the mediational model. Mainly, life satisfaction was predicted indirectly by career adaptability through courage. Such outcome has important implications for practice and highlights the need to support workers in planning their life design by developing career adaptability and workers' voluntary feeling to act, according to different levels of fear, when facing a threat to the achievement of a significant result or objective, which in turn will positively influence their feelings of life satisfaction.

**Keywords:** life satisfaction; courage; career adaptability; employees

## 1. Practioner Points

- Career adaptability positively contributes to the ability to face future challenges associated with difficulties regarding social and work uncertainties;
- Courage can enhance the value of an organization, increasing the levels of life satisfaction in workers;
- Ethics education and training in organizational settings concentrate on rigorous regulation and imposing compliance controls;
- Career counselors can help people develop their courage, set challenging goals, take calculated risks, support their desire for continuous learning and improved performance, and lead them to career success.

## 2. Introduction

The 2008 economic crisis exacerbated job-related problems; 2008 plunged the global economy into one of the most severe recessions in the industrial world. Furthermore, the COVID-19 pandemic exacerbated pre-existing economic challenges resulting from the 2008 crisis (Andersson and Arvidsson 2023). Many economies were still recovering from the lingering effects of the previous downturn, and the pandemic added new layers of complexity. Numerous international work organizations, such as the European Work Council, the International Labor Organization, and UNESCO, have emphasized the urgent need to undertake efficacious actions to create job opportunities and "better" jobs (International Labour Organization 2015). Promoting a social and solidarity economy may prevent increased loss of human well-being because of the dominant economic model. Scholars studying organizational ethics have underlined the importance of encouraging the development of moral strength in the workplace and maintained that this

would necessitate more than reinvented programs, policies, and penalties (Verschoor 2004; Wilmot and Ones 2019).

Instigating a behavioral revolution and encouraging a reintroduction of values, personal conscience, and responsibility may support this development (Gates 2004). Past scholars (Stansbury and Barry 2007; Weaver and Treviño 1999) have proposed that organizational ethics can be efficient when a value-based approach is mixed with typical compliance-based initiatives. However, although this concern has been recognized, organizations need to do more to cultivate the value-based approach.

Taking up the question posed by Aristotle (How do we live a good life?) and placing it in the employment context (How do we live a good job?), the person who lives in happiness strives to follow virtue, which contributes to the common good in some way (McAdams 2015). Positive organizational scholarship and positive psychology have tried to handle this issue by clarifying character strengths (Peterson and Seligman 2004) and virtue-based ethical performance (Cameron et al. 2003). Peterson and Seligman (2004) maintain that courageous actions are a combination of character strengths–which include persistence, integrity, bravery, and vitality–that promote "the exercise of will to accomplish goals in the face of opposition, either external or internal" (p. 199). Will-power helps face and settle ethical challenges and confront obstacles that may thwart the ability to take the right action (Sekerka and Bagozzi 2007). The long-term success of an organization can be due to acts of courage, which strongly impact employees (Kilmann et al. 2010).

The manuscript delves into a comprehensive exploration of courage in a group of Italian works, navigating through the theoretical approach of Life Design, one of the most accredited approaches in the field of vocational guidance. Subsequent sections delve into career adaptability, courage, and life satisfaction, presenting a synthesis of research findings to offer a holistic understanding of the subject. Considering the literature review, this study aims to test a mediational model to analyze the relationship between life satisfaction and career adaptability through courage.

The Life Design approach was created during the early stages of the financial crisis. It considers positive resources crucial to coping with significant pressures in terms of financial difficulty, decreased job offers, and increased career transition (Nota et al. 2015). The Life Design approach highlights skills and knowledge that can be applied to analyzing ecological settings, multiple contexts, non-linear causalities, and complex dynamics characterizing current career environments. Furthermore, Life Design emphasizes the need for help and support so that people can become experts in handling developmental tasks and career transitions (Savickas 2015). Career adaptability and positive resources toward work, such as courage, are considered the contemporary worker's essential resources to accomplish these goals, and thus capable of granting advantages involving the degree of personal and professional fulfillment perceived (Hartung 2015; Rossier 2015).

Considering said theory, this research is centered on career adaptability and courage, which may be crucial for workers' ability to overcome personal and professional obstacles and the degree of life satisfaction they perceive. Specifically, considering the Life Design paradigm within the context of career development, this study may be essential to support active people in designing their career path in line with personal values and life goals. The effort of this study is to integrate the concept of courage into the life design framework to sustain individuals in navigating the uncertainty of the contemporary work environment, which is often characterized by constant change. The courage to embrace uncertainty is essential for individuals to explore new opportunities with curiosity and openness to change. Furthermore, throughout career courage, individuals may face various transitions, such as changing roles and positions or pursuing further education. Courage becomes a catalyst for navigating these transitions. Life Design, coupled with the cultivation of courage, provides individuals with the tools to approach career changes proactively and confidently, and this could improve the perceived quality of life (Santilli et al. 2021).

In particular, this study aims to analyze the connection between life satisfaction, courage, and career adaptability in a group of employees.

- Career Adaptability. This variable comprehends a series of skills that favorably impact people's ability to handle their jobs or plan their lives (Rossier 2015). In particular, there are four resources at the center of said variable: (a) concern, the capability to plan the future by taking into consideration who the individual is and whom they wish to become; (b) control, the propensity to consider that the future can be manageable, albeit partially; (c) curiosity, a person's level of ability to explore the self (as regards knowledge, skills, abilities and values) and the environment; and (d) confidence, the belief that one can face challenges, and successfully cope with obstacles and barriers (Savickas 2011; Savickas and Porfeli 2012).

Career adaptability encompasses a series of regulatory procedures, or supporting regulation procedures, which are especially significant for the design of one's life and are implicated in the manifestation of career-related behaviors. In particular, it is crucial in supporting people planning their careers and private lives. Enhancing resilience and independence helps improve individuals' capability to accept and confront vagueness, insecurity, and professional and personal problems (Nota and Rossier 2015; Savickas 2011; Savickas and Porfeli 2012).

The adaptation model of career construction (Savickas 2013; Savickas and Porfeli 2012; Hirschi et al. 2015; Tolentino et al. 2014; Rudolph et al. 2017) suggests that people's career adaptability positively affects adapting responses, which in turn positively affects adaptation results. People's adaptive beliefs and behaviors in approaching career development tasks and fluctuating work and career conditions are signs of adapting responses (Hirschi et al. 2015; Savickas 2013; Savickas and Porfeli 2012). Adaptation results are corroborated by the person–environment goodness of fit and outcomes such as commitment, development, satisfaction, and work success (Savickas 2013; Savickas and Porfeli 2012).

Adapting responses correlate positively with promotion-focused self-regulation strategies, which imply being goal-focused, searching and finding alternative solutions, being proactive, and with task-oriented coping and cognitive reappraisal (van Vianen et al. 2012).

As regards adaptation results, career adaptability relates to workers' life satisfaction. Career adaptability was found to mediate, albeit partially, the relationship between, on one side, life satisfaction and, on the other, job insecurity and job strain in a group of Swiss adults, both employed and unemployed (Maggiori et al. 2013). Hirschi (2009) investigated how career adaptability can impact the development of a feeling of dominance and wellness as concerns enhanced life satisfaction since these are critical elements for the favorable development of young people.

- Courage. The opportunity to behave courageously often defines special moments of a person's career. This variable corresponds to "the ability to act for a meaningful (noble, good, or practical) cause, despite experiencing the fear associated with perceived threat exceeding the available resources" (Woodard 2004, p. 174). Courage implies a deliberate sensation to perform actions, according to different levels of fear, when facing a barrier to achieving a significant result or objective, maybe a moral one (Woodard and Pury 2007). Multiple components can be recognized in the definition of courage, among them (1) the existence of a threat and (2) a significant or worthy result. In the work context, Srivastva and Cooperrider (1998) have defined courage as a management virtue and described professional courage as a quality that prompts and empowers people to pursue the right pathway by the ethics of their profession (Harris 1999).

Aristotle defined courage as the first human virtue since, thanks to it, every other virtue becomes possible. Walston (2003) made a list of brave actions performed at work. Among its elements, we can find: (a) disclosing vulnerabilities, for example, when someone has to learn how to use a new software program and is not comfortable with this task because it creates feelings of anxiety; (b) expressing an unpopular opinion, this involves the fact to speak up about one's positions and judgments; and (c) dedicate oneself to reach long terms objectives, such as joining evening classes, renouncing to the present to emphasize

a future goal. Brave individuals express their objectives and work accordingly to reach them. They implement new procedures when previous strategies are no longer helpful, take chances and examine their objectives, and wonder if they would like to reach them, overcome obstacles, and create an unequivocal sight of their goals.

Different research studies have highlighted the correlation between career adaptability, positive variables, and life satisfaction in workers, even if there are no specific studies on courage. Considering courage as one of the other positive traits, Peterson et al. (2007) demonstrated that it robustly predicts life satisfaction, the cognitive component of subjective well-being (Pavot and Diener 1993). In their study, Kilmann et al. (2010) observed that members who perceive themselves as working in an organization in these acts of courage can be accomplished by recognizing less structural rigidity and control, more cultural support and direction, higher levels of perceived organizational performance, and higher satisfaction.

- Life Satisfaction. Life satisfaction is a crucial aspect of an individual's overall well-being (Pavot and Diener 1993), and the Life Design paradigm considers it an essential outcome in career development (Rossier 2015). Life satisfaction is defined by Diener (2000) as a cognitive process in which individuals evaluate the quality of their lives based on a set of established criteria.

Life satisfaction in workers can be determined as the overall sense of fulfillment and contentment that employees experience concerning their work, career, and personal life. It is closely linked to positive strengths such as courage, resilience, and optimism. Cultivating courage in the workplace can positively impact life satisfaction among workers (Porath et al. 2012). When employees feel empowered to take risks, make decisions, and confront challenges, they are more likely to experience greater satisfaction in their work and personal lives. Additionally, a culture that encourages and celebrates courage can foster a sense of purpose and meaning in the workplace, contributing to overall life satisfaction. Recognizing and promoting courage as a positive strength can significantly enhance the well-being and motivation of workers (Gao et al. 2019).

The relationship between career adaptability, courage, and life satisfaction is crucial for individuals to navigate their career paths successfully. By understanding the interplay between career adaptability, courage, and life satisfaction, individuals can make informed decisions and take proactive steps to design a fulfilling future despite uncertainties and challenges (Xu et al. 2017). Furthermore, having greater career adaptability resources can increase perceived career opportunities, the ability to cope with obstacles and challenges, and the possibility of successfully achieving career goals. All of these factors contribute to increased levels of life satisfaction (Kvasková et al. 2023). Findings also suggest that interventions to cultivate courage and enhance career adaptability can positively impact individuals' life satisfaction (Magnano et al. 2021). By courage, organizations and career counselors can support employees in navigating their career paths more effectively.

- *Research Aims*. Considering the Life Design approach, which underscores the importance of courage and career adaptability concerning professional challenges, this study aims to analyze the connection between life satisfaction, courage, and career adaptability in a group of employees. More specifically, a fully mediational model (Figure 1) between life satisfaction and career adaptability is tested through courage.

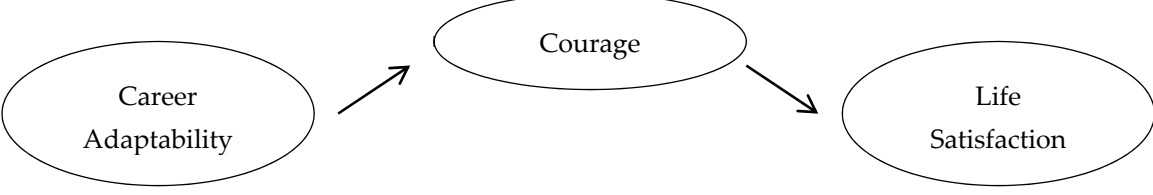

**Figure 1.** The mediated model is theoretically assumed.

### 3. Method

*3.1. Participants*

The study involved 525 employees, 291 men (55.4%) and 234 women (44.6%), with a mean age of 34.50 years (SD = 10.17). Men's mean age was 33.59 (SD = 9.24); women's mean age was 35.47 (SD = 11.04). Based on the Italian Statistic Institute work classification (ISTAT 2013), the types of work represented include legislators, business owners and senior management (5.6%); intellectual, scientific and highly specialized professions (23.9%); technical occupations (18.3%); executive work office professions (21.1%); trained occupations in commercial activities and services (5.6%); artisans, farmers and skilled workers (11.3%); system conductors, mobile machinery and vehicle drivers (2.8%); military (1.4%); and other unidentified professions (10%). Work experience ranged from 1 to 39 years; the average is 6.5 years.

*3.2. Measures*

Sociodemographic. The questionnaire collects a few demographic data, including age, gender, and work information, using open questions presented at the beginning of the quantitative measure.

Career Adapt-Abilities Scale-Italian Form (CAAS; Soresi et al. 2012). The instrument counts 24 items, analogously to the Career Adapt-Abilities Scale-International Form (Savickas and Porfeli 2012). People evaluated every item on a scale from 1 (Not strong) to 5 (Very Strong). The total score of the 24 items together shows career adaptability. Said items are separated into four subscales that assess the adapt-ability properties of concern (e.g., "Understanding that the options I choose today change my future"), control (e.g., "Depending on myself"), curiosity (e.g., "Exploring alternatives before choosing something"), and confidence (e.g., "Being conscious of one's abilities"). In the sample analyzed, Cronbach's alpha for the four subscales is 0.79 (concern), 0.72 (control), 0.78 (curiosity), and 0.81 (confidence). Cronbach's alpha for the total score is 0.90.

The Courage Measure (CM; Norton and Weiss 2009). The measure includes 12 items developed to explore a functional definition of courage, "the persistence or perseverance despite being afraid". The Likert scale was 7-point, from 1 (Never) to 7 (Always). Some instances of items are: "I am prone to confront my fears"; "Despite feeling very frightened, I would remain in that circumstance until I have done what I have to do".

In a study conducted to adapt and validate the Italian version of the scale in a group of adults, the researchers Ginevra et al. (2017) marked two factors accounting for 70% of the variance: courage measures (e.g., "I tend to face my fears") and avoidance measures (e.g., "If the thought of something makes me anxious, I will usually avoid it"). Cronbach's alpha was 0.81 (courage measures) and 0.60 (avoidance measures). The Courage subscale correlated positively with adults' future orientation, resilience, optimism, hope, and life satisfaction. For this sample, the courage subscales reported a Cronbach's alpha of 0.88.

The "Satisfaction with Life" Scale (Diener et al. 1985). It is composed of 5 items that assess overall life satisfaction. The range of the total scores is from 5 to 25. The higher scores indicate increased life satisfaction. An instance of an item is "My life pleases me". In research carried out to adjust and confirm the Italian version of the scale in a group of parents, Ginevra et al. (2017) obtained a mono-factorial structure, which accounted for 55.73% of the total variance, and a Cronbach's alpha of 0.85. In this study, Cronbach's alpha is 0.88.

*3.3. Procedure*

A multidisciplinary team interested in exploring the topic of courage launched a research program at an Italian University. The study participants were employees who were part of the said research program. All employees (n = 550) were assured of the confidentiality of the research and were invited to fill out a web-based survey. Among the employees, 525 agreed to participate and filled out the survey (a response rate of 95.63%).

The respondents were informed that the study aimed to investigate several issues related to work and life.

Participants were also informed that, once the data had been processed, they would be individually presented with a personalized report on their results. Individual counseling was available to those who asked for it. Administration lasted about 20 min.

### 3.4. Data Analysis

Preliminary analysis. First, there has been the computation of means, standard deviations, and inter-correlations. To verify if there were any critical across-group gender differences in courage, career adaptability, and life satisfaction, three preliminary T-tests were carried out.

Mediational analysis. The model hypothesized by researchers was tested through structural equation modeling, SEM (MPLUS 7, Muthen and Muthen 2014). SEM was used to test the meditational effects of courage.

A single measure variable expressed the courage and life satisfaction constructs implemented in this research. Therefore, we followed the guidelines of Little et al. (2002) to produce item parcels for latent constructs, and, using a specific balancing technique (item-to-construct; Little et al. 2002), we created two parcels for life satisfaction and three for courage.

Regarding career adaptability, item parcels were generated to form multiple manifest indicators representing each latent construct. As proposed by Kishton and Widaman (1994), we decided to use the internal consistency approach, which consists of creating parcels that use factors as grouping criteria. So, four indicator variables were generated in line with Savickas and Porfeli's (2012) theoretically and empirically validated subscales (concern, control, curiosity, and confidence).

To examine the model fit, we used the maximum-likelihood (ML) estimation method (Quintana and Maxwell 1999). We chose the chi-square test, which is by far the most commonly used goodness-of-fit index (Quintana and Maxwell 1999). Since this test is highly influenced by model complexity and sample size, we considered other indices to evaluate the model's better fit. The comparative fit index (CFI) confronts the model we theorized with the null model to look for improvements. The CFI varies from 0 to 1, with a value close to 1 indicating an outstanding fit and values greater than 0.90 indicating an acceptable fit (Bentler 1990). To test the fitting of the model, we decided to use the root-mean-square error of approximation (RMSEA) because it is not affected as much as the chi-square by a small sample size. RMSEA values lower than 0.05 point to a good fit, and RMSEA higher than 0.08 means approximation errors (Hu and Bentler 1999). We used the standardized root-mean-square residual (SRMR) because it tests the overall difference between observed and predicted correlations. Values of SRMR lower than 0.10 are typically considered favorable (Hu and Bentler 1999).

In line with Hoyle and Panter (1995), we tried to establish if the theorized model was better than other competing models. Therefore, we compared couples of tested models using chi-square difference tests. Following Shrout and Bolger's (2002) suggestions, we used the bootstrapping procedure to test the magnitude and significance of mediation effects. Therefore, we formed 5000 bootstrap samples from the original data set through random sampling with replacement. The authors suggested that researchers report a 95% confidence interval (CI) for the mean indirect effect. A 0.05 level is considered statistically significant for the indirect effect if zero is not included in the CI.

## 4. Results

### 4.1. Preliminary Analysis

Table 1 summarizes means, standard deviations, and inter-correlations. The correlations observed between career life satisfaction, adaptability, and courage, and between courage and life satisfaction, were moderate and positive (see Table 1).

Moreover, the variance inflation factors (VIFs) were checked to test the model and resulted in an equal to 1.00, much lower than the recommended 5.0 (Hair et al. 2011), excluding multi-collinearity.

Three T-tests did not reveal any significant gender differences concerning career adaptability, $t(523) = 0.488$, $p = 0.485$, courage $t(523) = 0.3.812$, $p = 0.052$, and life satisfaction $t(523) = 1.526$, $p = 0.217$ (see Table 1).

**Table 1.** Means, standard deviations, and intercorrelations.

| | 2 | 3 | Men M | Men SD | Women M | Women SD |
|---|---|---|---|---|---|---|
| 1. Career adaptability | 0.460 ** | 0.254 ** | 93.65 | 12.53 | 95.29 | 12.44 |
| 2. Courage | | 0.191 ** | 38.19 | 7.88 | 38.68 | 8.72 |
| 3. Life satisfaction | | | 22.73 | 5.77 | 22.05 | 6.20 |

Note: ** $p < 0.001$.

### 4.2. Measurement Model

The hypothesized measurement model showed a good fit with the observed data, $\chi^2 (24) = 34.620$, CFI = 0.99, TLI = 0.99, RMSEA = 0.03, SRMR = 0.02. All the standardized path estimates of the manifest indicators (ranging from 0.61 to 0.94) revealed statistical significance.

### 4.3. Structural Model

A good fit to the data was supported by the results for the partially mediated model (Model A), $\chi^2 (24) = 30.962$, CFI = 0.99, TLI = 0.99, RMSEA = 0.03, SRMR = 0.03.

The fit indices of the full structural model were adequate. This is supported by the values of data $\chi^2 (25) = 32.336$, CFI = 0.99, TLI = 0.99, RMSEA = 0.03, SRMR = 0.04.

Model A (partially mediated model) did not present a marginally better fit to the data than Model B (fully mediated model), $\Delta\chi^2(1) = 1.374$, $p = 0.24$ (see Figure 2). The bootstrapping analyses highlighted that career adaptability (ß = 0.15, $p < 0.01$ [0.035, 0.270]) had a significant indirect connection (i.e., it did not include zero) with life satisfaction through the mediating role of courage.

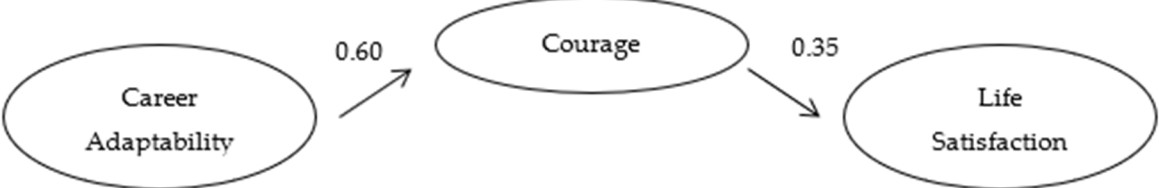

**Figure 2.** Significant standardized prameter estimates in the partially mediated model.

## 5. Discussion

Based on the Life Design approach, the present study examined the relationship between life satisfaction, courage, and career adaptability in a group of Italian workers. More precisely, we theorized that courage indirectly mediated the relationship between life satisfaction and career adaptability. Our results corroborated such mediation, showing that, through courage, life satisfaction is linked to career adaptability.

People's adaptive beliefs and behaviors in approaching career development tasks and fluctuating work and career conditions are signs of adapting responses (Hirschi et al. 2015; Savickas 2013; Savickas and Porfeli 2012). Adaptation results are corroborated by the person–environment goodness of fit and outcomes such as commitment, development, satisfaction, and work success (Savickas 2013; Savickas and Porfeli 2012).

Results obtained are consistent with other studies showing that career adaptability positively contributes to the ability to face future challenges associated with difficulties

regarding social and work uncertainties. Specifically, they align with the career construction adaptation model (Savickas 2013; Savickas and Porfeli 2012; Hirschi et al. 2015; Tolentino et al. 2014). In the present study, it has emerged that people's career adaptability positively influences adapting responses, workers' tendency to act according to different levels of fear to respond to a threat to attain a significant goal or objective, which in turn positively influences adaptation results, that is higher feelings of life satisfaction (Kilmann et al. 2010).

Furthermore, our results highlight that courage can enhance an organization's value by increasing employee satisfaction. These results lead to the organization developing the ability to teach managers and workers to engage in virtue excellence and focus on career transition with courage. When a different pattern of behavior is required, a different approach to ethics must be contemplated, cultivating moral strength (Sekerka et al. 2009).

Implication for Theory. The theoretical implications of the study's findings, rooted in the Life Design approach, are multi-faceted and contribute to our understanding of the interplay between life satisfaction, courage, and career adaptability. First, the results validate and align with existing research on career adaptability. The study further supports the notion that individuals with higher levels of career adaptability exhibit a positive orientation toward dealing with challenges in a dynamic work environment (Santilli et al. 2021). About the mediating role of courage as an indirect mediator between life satisfaction and career adaptability, the present contribution expands on previous frameworks that may have yet to explicitly consider the role of courage in connecting life satisfaction to adaptive career behaviors (Magnano et al. 2021).

Furthermore, the findings underscore the organizational value of cultivating courage in workers that may witness increased life satisfaction among employees, potentially leading to positive outcomes for the organization. The study highlights the need for ethical considerations in career transitions. Cultivating moral strength and ethical behavior becomes essential, especially when individuals must exhibit courage in pursuing significant career goals (Nota et al. 2015).

Implications for Practice. Our results show that increasing workers' career adaptability resources in activities and interventions could be of great advantage, including the development of each of the adaptability resources, i.e., concern for a hopeful vision of one's future; control to increase internal locus of control and decision-making skills; curiosity to stimulate career exploration; confidence and self-efficacy beliefs to encourage agency and coping skills (Nota and Rossier 2015). Although building moral strength is very important to cultivate a different culture of organization, regrettably, ethics education and training in organizational settings keep concentrating on rigorous regulation and imposing compliance controls. Career counsellors can help people develop their courage, set challenging goals and take calculated risks, support their desire for continuous learning and improved performance, and lead them down the path to career success (Walston 2003).

Limitations and Future Directions. The present study shows some limitations. First, future studies should involve more participants to check the generalizability of the suggestions formulated here. In addition, more information regarding work contracts should be considered, such as moderators of the relationship between courage, career adaptability, and life satisfaction. Another limitation is that only self-evaluation questionnaires were used. Future research should explore and extend these findings by using quantitative and qualitative data and considering leadership evaluation. Future research should also consider other positive psychological variables, such as hope and optimism, and other less subjective dimensions of quality of life. One last suggestion for future research is to use a longitudinal design to verify career adaptability and whether these characteristics can help workers, over time, in career transitions and retirement.

## 6. Conclusions

In conclusion, the findings of this study, conducted within the framework of the Life Design approach, have shed light on the intricate interplay between life satisfaction, courage, and career adaptability among Italian workers. The investigation revealed that

courage is an indirect mediator, linking life satisfaction to career adaptability. The study's outcomes are consistent with prior research highlighting the positive contribution of career adaptability to individuals' ability to confront challenges associated with social and work uncertainties. Notably, the study contributes to understanding the legislative implications of courage. The findings suggest that cultivating courage in workers can elevate organizational value by increasing overall life satisfaction. This insight implies a potential for organizations to teach and encourage virtue excellence, emphasizing the importance of courage in navigating career transitions. Practically, the study underscores the importance of enhancing workers' career adaptability resources through targeted activities and interventions. Career counsellors can play a vital role in supporting individuals to develop courage, set challenging goals, and foster continuous learning—a pathway to achieving career success.

**Author Contributions:** Conceptualization, S.S.; methodology, S.S. and M.C.G.; data curation, S.S. and M.C.G.; writing—original draft preparation, S.S., M.C.G. and I.V.; writing—review and editing, I.V. and B.R.; supervision, M.C.G.; project administration, S.S. All authors have read and agreed to the published version of the manuscript.

**Funding:** This research received no external funding.

**Institutional Review Board Statement:** All phases of this study were conducted according to the Ethical Code for Italian Psychologists (L. 56/1989) and Legislative Decree Data Privacy and the Protection of Personal Data (LD 196/2003). Moreover, this research was designed and implemented respecting the Ethical Standards of the Italian Association of Career Guidance (SIO) and the Italian Association of Psychologists (March 2015, revised in 2022). Specifically, the ethical code of the Italian Association of Psychology, approved in March 2015 and revised in July 2022, draws inspiration from the Declarations of Helsinki (1964/2013) and emphasizes that about psychological research with human beings, the Ethics Committee pay special attention to researches involving (a) a risk to the psychological and physical well-being of persons participants; (b) the participation of vulnerable people (such as minors, persons unable to express consent, imprisoned persons, hospitalized or institutionalized persons; groups exposed to stigma or risk of social discrimina tion); (c) the use of biomedical apparatus and invasive investigative tools; (d) the use of deception; (e) the use of stimuli that may hurt the personal and cultural sensitivities of the persons participating; (f) the introduction of limitations on the right to anonymity and confidentiality of participants. This study does not fall into any of these cases. However, in conducting this research we respected all rules of conduct under the code of ethics. Specifically, participants were informed about the confidentiality of the research, why it was conducted, how their data would be used and if there were any risks associated (article 1). They were also informed that once the data had been processed, they would be individually presented with a personalized report on their results and that nobody could read their answers without their consent (articles 3 and 4). Finally, participants were assured of the anonymity of the research (article 4). All subjects gave their informed consent for inclusion before they participated in the study.

**Informed Consent Statement:** Informed consent was obtained from all subjects involved in the study.

**Data Availability Statement:** The data presented in this study are available on request from the corresponding authors.

**Conflicts of Interest:** The authors declare no conflicts of interest.

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
