# Peer review of "Life Satisfaction, Courage, and Career Adaptability in a Group of Italian Workers"

_socsci, doi:10.3390/socsci13020106_

Round 1

Reviewer 1 Report

Comments and Suggestions for Authors

Thanks for allowing me to review the manuscript "Life Satisfaction, Courage, and Career Adaptability in a Group of Italian Workers." I appreciate the topic and the study conducted. I think the manuscript would benefit from a few minor structure revisions. For example, the authors could consider presenting the aim of the study earlier in the introduction and following up with the rest of the introduction, in which they present the considered variables. Moreover, the manuscript is quite long, and I would suggest to include an overview paragraph in which you can present the contents of the manuscript. This can be done as follows: "In the rest of the paper, we firstly present...,Second, we continue with ... We conclude". I make this suggestion in order to improve the manuscript for potential readers in mind.

In respect to the methodology, I would suggest the authors to specify the demographic variables included in the study. How did they measure gender? What about work information?

Thanks for your work

Comments on the Quality of English Language

Good English 

Author Response

Dear Reviewer, 

thanks for the review and for the time spent in giving us feedback to improve the manuscript. 

Following reviewers’ suggestions, we consider all of them and insert all the changes in red throughout the text. 

Based on the revision suggested, we modified the following parts:

  • Introduction: we improved the clarity and the straightness of this section, inserting an overview paragraph with the contents of the manuscript (lines 63-69); adding a paragraph about the construct of life satisfaction (lines 166-180), as well as a paragraph about the relationship between the three constructs (lines 181-192). We inserted the aim of the study presented earlier in the introduction, specifying the hypothesis and the significance of this study within the field of career development (lines 84-98). A figure explaining the research model was also inserted (lines 200-203)
  • Moreover, considering the first reviewer, we updated the economic background in light of the effects of the Covid crisis and post-Covid conditions on the labour market (lines 30-34). We also updated the literature by inserting different new references such as:
  • Saleem, R., Azam, S., & Hashmi, A. H. (2024). COVID-19 and Its Global Economic Effects: Erstwhile and Afterwards. In Post-Pandemic Economy, Technology, and Innovation (pp. 1-47). Apple Academic Press.
  • Ginevra, M. C., Magnano, P., Lodi, E., Annovazzi, C., Camussi, E., Patrizi, P., & Nota, L. (2018). The role of career adaptability and courage on life satisfaction in adolescence. Journal of adolescence, 62, 1-8.
  • Molina, O., & Godino, A. 2021. Scars that Never Heal: Dualisation and Youth Employment Policies in Spain from the Great Recession to the Corona Crisis. Scars that Never Heal: Dualisation and Youth Employment Policies in Spain from the Great Recession to the Corona Crisis, 111-132.

As regards the methodology section:

  • We clarify How we measured the socio-demographic variables specified (Measure section, lines 217-219) a deeper explanation of the methodological procedure for data analysis was provided by the authors (Data analysis section, lines 295-303). Specifically, we used mediation modelling with bootstrapping, which is a statistical technique used to examine the indirect effect or mediation of an independent variable on a dependent variable through one or more intermediate variables. Baron and Kenny (1986) proposed one of the early mediation analysis procedures, while we use Shrout and Bolger's methodology (2002) which is associated with procedures related to assessing the significance of mediation effects using bootstrapping. Furthermore, we tested the hypothesized measurement model that showed a good fit with the observed data, χ2 (24) = 34.620, CFI = .99, TLI = .99, RMSEA = .03, SRMR = .02. All the standardized path estimates of the manifest indicators (ranging from .61 to .94) revealed statistical significance.

As regards this last point, please note that we used mediation modelling with bootstrapping, which is a statistical technique used to examine the indirect effect or mediation of an independent variable on a dependent variable through one or more intermediate variables. Baron and Kenny (1986) proposed one of the early mediation analysis procedures, while we used Shrout and Bolger’s methodology (2002) which is associated with procedures related to assessing the significance of mediation effects using bootstrapping. 

Discussion:

  • We improved the Interpretation of the result including the Practical and theoretical implication (lines 364-379) and including a conclusion paragraph (lines 405-419).

Finally, we updated the literature throughout the text of the manuscript and a revision related to APA 7 guidelines.

Reviewer 2 Report

Comments and Suggestions for Authors

Thank you for the opportunity to review this original article. 

The introduction is comprehensive and introduces the issue. I have my doubts that the economic background should be limited only to the 2008 crisis. Perhaps it is worth outlining the panorama of the labor market in the COVID-19 crisis and post-COVID conditions? The literature analysis is of a good standard. However, the number of cited articles could be higher, such as:

- Ginevra, M. C., Magnano, P., Lodi, E., Annovazzi, C., Camussi, E., Patrizi, P., & Nota, L. (2018). The role of career adaptability and courage on life satisfaction in adolescence. Journal of adolescence, 62, 1-8.

- Magnano, P., Lodi, E., Zammitti, A., & Patrizi, P. (2021). Courage, career adaptability, and readiness as resources to improve well-being during the University-to-Work Transition in Italy. International Journal of Environmental Research and Public Health, 18(6), 2919.

- Parola, A., Zammitti, A., & Marcionetti, J. (2023). Career Calling, Courage, Flourishing and Satisfaction with Life in Italian University Students. Behavioral Sciences, 13(4), 345.

I also suggest formulating research hypotheses and placing them in the figure with the research model.

The research procedure is not objectionable. The questionnaires used are tested in other studies and adapted to Italian conditions.

The presentation of the results discussion and conclusions are correct. The research oranities have not been forgotten.

Author Response

(The authors gave the same response as above.)

Reviewer 3 Report

Comments and Suggestions for Authors

 Dear authors

It was with great pleasure that I reviewed your manuscript.

However, I have a few comments to make:

1. Most of the bibliography needs to be updated. It needs to be updated;

2. I have not seen a literature review on the construct of "life satisfaction". It needs to be done.

3. There is no literature review on the relationship between the three constructs to support your hypothesis(es). It would be best if you did it.

4. I need help finding your hypothesis (es) throughout the manuscript. It would be best if you put the hypotheses in as you discuss the constructs' relationship. 

5. At the end of the introduction, you should put the research model in order to make the manuscript more accessible to read.

6. You should carry out confirmatory factor analyses of each instrument used and report the respective fit indices.

7. Based on the confirmatory factor analyses, they should calculate construct reliability, convergent and divergent validity.

8. Since this is a mediating effect, they should have tested the three conditions mentioned by Baron and Kenny (1986).

9. You have yet to include the practical and theoretical implications in the discussion.

10. The conclusions chapter needs to be included.

My Best Regards

Author Response

(The authors gave the same response as above.)

Reviewer 4 Report

Comments and Suggestions for Authors

The article delves into the intricate relationships between life satisfaction, courage, and career adaptability among Italian workers within the Life Design framework. Employing robust statistical methods, including structural equation modeling, the study establishes certain connections among these variables. However, significant weaknesses hamper its potential contribution to the field.

Introduction Review:

The introduction sets the stage by integrating diverse elements related to economic crises, job concerns, and the imperative for ethical shifts in organizations. Nonetheless, several areas need refinement:

- Clarity and Structure: The introduction lacks a clear structure, making it somewhat convoluted and challenging to follow. There's a need for better organization and segmentation of ideas to enhance readability. The article lacks context, failing to provide an adequate introduction to the significance of the study within the broader field of organizational psychology or career development.

- Citation Integration: Better integration of references into the narrative is needed for coherence. While numerous sources are mentioned, the integration of these citations into the narrative could be improved. Rather than listing studies one after another, weaving these references more coherently into the text would enhance the flow and make the argument more persuasive.

- Conciseness: Some sections feel excessively detailed, potentially diluting the core message. Streamlining the content by focusing on the most crucial points could make it more concise and impactful.

- Research Gap Clarification: Explicitly stating the research gap would accentuate the study's significance. The introduction touches upon various theoretical frameworks and their implications for workers' well-being. However, it could explicitly state the gap in existing research that this study aims to address. This would help in establishing the novelty and significance of the forthcoming research. The article lacks an in-depth theoretical grounding despite referencing the Life Design approach. There's a need for a more profound theoretical foundation to underpin the relationships between life satisfaction, courage, and career adaptability.

- Synthesis of Ideas: A more cohesive synthesis of theories would strengthen the argument. While the introduction covers a wide range of theoretical underpinnings, it would benefit from a more cohesive synthesis of these theories to build a stronger argument for the research study.

- Research Objectives: Sharper and more specific research objectives are needed for clarity. Although the research aims are broadly outlined, providing a more specific and concise statement of the research objectives would give readers a clearer understanding of the study's intended outcomes.

Discussion Section Review:

- Lack of Depth in Interpretation: Like the Results section, the discussion lacks depth in interpreting findings. It could benefit from a more thorough analysis of how and why relationships exist, delving deeper into theoretical underpinnings.

- Vague Statements and Lack of Specificity: Some statements are broad or lacking specificity, requiring more elaboration or practical examples to strengthen the discussion's impact.

- Limited Exploration of Limitations and Future Directions: While limitations and future directions are mentioned, there's a need for a more detailed discussion on how limitations may have influenced results and a clearer rationale behind future research suggestions. The article mentions potential areas for future research but fails to provide a clear rationale or direction for how addressing these areas could advance the field or address the study's limitations.

Improvement in these areas - through clearer organization, in-depth theoretical underpinning, more precise discussions, and better integration of literature - would significantly enhance the article's impact and contribution to the field.

Author Response

(The authors gave the same response as above.)

Round 2

Reviewer 3 Report

Comments and Suggestions for Authors

Dear Authors

Thank you for taking on board my suggestions for improvement.

My Best Regards

Reviewer 4 Report

Comments and Suggestions for Authors

I congratulate the authors for the improvements to the article, which are visible and enrich the work.